# Nanoparticle-Based Immunotherapy for Reversing T-Cell Exhaustion

**DOI:** 10.3390/ijms25031396

**Published:** 2024-01-23

**Authors:** Fei Li, Yahong Wang, Dandan Chen, Yunjie Du

**Affiliations:** 1Institute of Pathogen Biology, School of Basic Medical Sciences, Lanzhou University, Lanzhou 730000, China; 120220905730@lzu.edu.cn; 2School of Public Health, Lanzhou University, Lanzhou 730000, China; wyahong2023@lzu.edu.cn (Y.W.); 220220912200@lzu.edu.cn (D.C.)

**Keywords:** T-cell exhaustion, nanoparticle, immune checkpoint blockade, tumor microenvironment, T cell metabolism

## Abstract

T-cell exhaustion refers to a state of T-cell dysfunction commonly observed in chronic infections and cancer. Immune checkpoint molecules blockading using PD-1 and TIM-3 antibodies have shown promising results in reversing exhaustion, but this approach has several limitations. The treatment of T-cell exhaustion is still facing great challenges, making it imperative to explore new therapeutic strategies. With the development of nanotechnology, nanoparticles have successfully been applied as drug carriers and delivery systems in the treatment of cancer and infectious diseases. Furthermore, nanoparticle-based immunotherapy has emerged as a crucial approach to reverse exhaustion. Here, we have compiled the latest advances in T-cell exhaustion, with a particular focus on the characteristics of exhaustion that can be targeted. Additionally, the emerging nanoparticle-based delivery systems were also reviewed. Moreover, we have discussed, in detail, nanoparticle-based immunotherapies that aim to reverse exhaustion, including targeting immune checkpoint blockades, remodeling the tumor microenvironment, and targeting the metabolism of exhausted T cells, etc. These data could aid in comprehending the immunopathogenesis of exhaustion and accomplishing the objective of preventing and treating chronic diseases or cancer.

## 1. Introduction

During acute infections, naive T cells are activated and undergo differentiation into effector T cells and memory T cells [1,2,3]. Effector T cells play a role in eliminating antigens and controlling infections. Following antigen clearance, the majority of effector cells would die by apoptosis [4]. Only about 5–10% of T cells persist and continue to differentiate into memory T cell subsets [5]. These memory T cells have long-term survival and retain the capability of homeostatic proliferation [6,7]. When re-stimulated, memory T cells can generate effector T cells and sustain a recall response [8].

In contrast, during several chronic infections and cancer, due to persistent antigen stimulation, antigen-specific T cells become dysfunctional or even exhausted, which is characterized by a decreased effector function, the sustained expression of various inhibitory receptors, such as PD-1, TIM-3, and cytotoxic T lymphocyte antigen 4 (CTLA-4), and a loss of memory ability [9,10,11]. T-cell exhaustion usually leads to disease progression. Recently, targeted immune checkpoint molecules have been widely studied and can effectively reverse T-cell exhaustion [10]. Although immune checkpoint blockade therapy shows great promise, there are still limitations, such as limited response rates, the possibility of relapse, and toxicity [12,13]. Therefore, it is necessary to summarize the characteristics of exhaustion to explore new therapeutic targets for reversing T-cell exhaustion in chronic infections and cancer.

Nanoparticle (NP)-based delivery systems show promise to act as drug carriers and can enhance the efficiency of antigen delivery [14]. Besides enhanced treatment effectiveness and decreased side effects, these systems possess great potential in immunotherapy due to their targeting abilities and stimulation-responsive properties [15,16]. In addition, combining NPs that target immune checkpoints with multiple therapeutic approaches has been shown to have a better effect on reversing exhaustion. In this review, we summarize the characteristics of exhausted T cells and discuss recent progress in NP-based immunotherapy for exhaustion therapy. Finally, we provide a summary on the existing problems and discuss future challenges and perspectives of nanoparticle application and exhaustion.

## 2. T-Cell Exhaustion Is Common in Both Infectious Diseases and Cancer

Immune dysfunction that occurs following persistent viral and bacterial infections poses a threat to human health. During chronic pathogen infections, the immune system is unable to quickly eliminate antigens, causing them to persist in the body, resulting in T-cell dysfunction or even exhaustion [17]. T-cell exhaustion is initially observed in the chronic lymphocytic choriomeningitis virus (LCMV) infection [18], as well as in cancer and other chronic infections like the hepatitis B virus (HBV), human immunodeficiency virus (HIV), and *Mycobacterium tuberculosis* (*M. tuberculosis*) [9,10,19]. The characteristics of exhausted T cells are a gradual loss of effector function, the excessive expression of multiple inhibitory checkpoints, and alterations in transcriptional programming [11]. Continuous antigen stimulation, hypoxia, and high reactive oxygen species (ROS) levels are the main factors that drive exhaustion [20]. When the immune system becomes “exhausted”, it becomes incapable of effectively resisting the invasion of foreign pathogens, thus losing its ability to eliminate pathogens [21].

When exhaustion occurs, T cell phenotypes change. Exhausted T cells overexpress multiple cell surface inhibitory immune checkpoints such as PD-1, TIM-3, CTLA-4, and lymphocyte activation gene 3 (LAG-3) [22]. In addition, recent studies have found that an immune regulatory molecule, CD39, is also highly expressed on the surface of exhausted T cells. CD39 is a surface-expressed ATP ecto-nucleotidase and is utilized to define exhaustion [23,24]. In general, the more inhibitory checkpoints co-expressed on exhausted T cells, the more severe the exhaustion [11].

T-cell exhaustion is a process of progressively losing their function [25], starting with a loss of cytotoxicity, proliferation potential, and IL-2 secretion, followed by a loss of IFN-γ and TNF-α production, ultimately impairing the ability to confer protection [10,11]. Furthermore, the expression of transcription factors also changes. For instance, the upregulated expression of B lymphocyte-induced maturation protein-1 (Blimp-1) [26] and the nuclear factor of activated T cells (NFAT) [20,27] and the downregulated expression of the T-box-containing protein expressed in T cells (T-bet) [28].

Exhausted T cells are heterogeneous in both phenotype and function [29], and can be classified into two main clusters: progenitor exhausted and terminally exhausted T cells. Progenitor exhausted subsets refer to a population of exhausted T cells that are similar to stem cells but express PD-1 and T-cell factor 1 (TCF-1) [9,30]. This subset has the ability to self-renew and proliferate and shows a good blocking response to the PD-1/PD-L1 pathway [31,32]. In contrast, terminally exhausted T cells exhibit an impaired proliferation ability and they have no response to PD-1 pathway blocking [33]. They have high expression of PD-1 and TIM-3 and a loss of TCF-1 expression [34]. Emerging insight redefines the phenotypic diversity of later-stage exhausted T cells, including terminal exhaustion and a cytotoxic phenotype expressing the killer cell lectin-like receptor [35]. In this article, “exhausted T cells” primarily refers to T cells that are terminally exhausted.

These two types of exhausted cells also exhibit different metabolic characteristics. Progenitor exhausted T cells manifest a catabolic metabolism and mainly utilize mitochondrial fatty acid β-oxidation (FAO) and oxidative phosphorylation (OXPHOS) as sources of fuel energy [36]. Conversely, terminally exhausted T cells primarily depend on glycolysis, with a reduced mitochondrial OXPHOS metabolism and decreased glycolysis [37,38,39,40,41]. In addition, terminally exhausted T cells show a reduced PGC-1α transcription and expression, which is involved in controlling mitochondrial biogenesis [38,39]. In terminally exhausted T cells, mitochondrial dysfunction is mainly manifested by an increased mitochondrial mass and reduced mitochondrial membrane potential, making it impossible for cells to effectively utilize OXPHOS for energy production [37,42]. The impaired mitochondrial OXPHOS restricts T cell proliferation and effector function by limiting ATP synthesis [43].

Targeting immune checkpoints has been extensively described as an efficient way to restore immunity and reinvigorate exhaustion [44]. Additionally, blocking the ligands of immune checkpoint molecules also achieves effective immunotherapy against tumor-induced exhaustion. For instance, Galectin-9 is a TIM-3 ligand that acts as a negative regulator and can induce cell death in the tumor microenvironment [45]. It has been found that blocking Galectin-9 can induce anti-tumor immunity and reverse the exhaustion of effector T cells [45,46]. Furthermore, multi-antibody combination therapy has demonstrated significant efficacy in rejuvenating exhausted T cells, but the reinvigoration remains incomplete and still has numerous limitations [47,48]. For example, the use of antibodies often requires high doses and long-term usage [49]. Additionally, aside from the limited response rates and toxicity, relapse is frequent, and many forms of cancer do not react to a single immune checkpoint blockade [12,13]. Consequently, it is imperative to explore the novel pathogenesis of exhaustion and explore new therapeutic targets to combat T cell exhaustion.

## 3. Nanoparticle Classification and Application

Nanoparticle (NP)-based delivery approaches can reduce side effects and toxicity in non-target cells in immunotherapy, thereby significantly improving the effectiveness of immunotherapy [50]. NPs have remarkable features, such as adjustable structures, a strong biomolecular loading capacity, abundant surface modification, and controllable release molecules [51,52]. NP-based delivery systems provide extended circulation and active targeting [53,54]. They can target solid tumors by targeting tumor cells, stimulating or reprogramming immune cells, remodeling the tumor microenvironment, and altering immune responses, thereby generating effective antitumor immunity [55]. Moreover, NPs can be easily modified to bind to specific receptors or ligands, thereby enhancing compatibility and efficiency [56]. As a result, NP-based strategies have gained widespread attention in disease treatment.

With the advancement of nanotechnology and materials science, numerous types of NPs have been developed and applied in delivery systems [57]. Nanomaterials primarily consist of organic nanomaterials (such as polymers and lipids), inorganic nanomaterials (such as metals, oxides, and carbon), and hybrid nanomaterials (such as lipid polymers and metal organic) [58,59]. These NPs can be designed and functionalized based on the properties and requirements of different drugs or biomolecules, enabling efficient and safe delivery in vitro. Consequently, lipid, polymeric, and inorganic NPs are engineered and applied to enhance precision therapies [57].

### 3.1. Lipid-Based NPs

Lipid-based NPs are the most common ones with high safety [60,61]. Lipid-based NPs are typically composed of phospholipids, ionizable lipids, cholesterol, and PEGylated lipids [15]. The most typical form of these NPs is spherical particles, which primarily consist of an internal hydrophilic part and an external lipid molecular layer. Lipid-based NPs as a delivery system have numerous advantages, including self-assembly, simple emulation, the ability to carry large loads, biocompatibility, and adjustable physicochemical properties [62]. The most typical representatives of such NPs are liposomes and lipid NPs (LNPs).

Liposomes, the most numerous types, are nanoparticle vesicles formed by the self-assembly of amphiphilic phospholipid molecules. They have been utilized in various scientific fields, typically for loading and delivering compounds with various properties, such as lipophilic, amphiphilic, or hydrophilic compounds [63]. Liposomes are frequently employed as carriers for gene delivery, enabling the encapsulation of DNA or RNA for gene therapy or gene editing. Due to the rapid absorption of liposomes by the reticuloendothelial system, their application is limited. Typically, the surface modification of liposomes is performed to prolong their circulation and enhance delivery, enabling their clinical use [64]. Additionally, the size of NPs affects cellular uptake. Generally, NPs smaller than 10 nm in diameter are rapidly cleared by the kidneys, while NPs with a diameter exceeding 200 nm are prone to triggering the complement system. Hence, the typical size of NPs used in immunotherapy is usually between 10 and 200 nm [65].

Furthermore, another type of lipid-based NP is LNPs. Recently developed LNPs are mainly composed of ionizable lipids, helper lipids, PEG-lipids, and cholesterol [66,67]. LNPs are extensively used for delivering nucleic acids and resemble liposomes in structure. However, the main distinction lies in the micelle structure formed within the core of LNPs, which can be altered according to the formulation and synthesis parameters [68]. Due to their effectiveness in delivering nucleic acid, with the advantages of a small size, simple synthesis, and serum stability, LNPs play a crucial role in personalized gene therapy applications [69,70]. However, the disadvantages of the LNPs system include a low drug load and limited biological distribution, leading to a high uptake in the liver and spleen [61].

### 3.2. Polymeric NPs

Polymer NPs are another important class of nanoparticle carriers, which have a variety of compositions and forms [57]. Polymer NPs also possess flexible and controllable delivery capabilities. The therapeutic agent can be encapsulated inside the NPs, encased in the matrix, or chemically coupled to the NPs surface or with the polymer. With this feature, polymer nanoparticles can effectively carry a variety of materials, such as drugs, biomacromolecules, various proteins, and vaccines [71]. Although the transfection efficiency of polymer NPs is relatively low compared with lipid-based NPs, the structure of polymers is more stable and easier to modify. By introducing functional groups such as thiol groups, polymers NPs can respond to stimuli such as ROS, pH, and enzymes. The modification of ligands such as targeting peptides and antibodies promotes the specific targeting of the delivery system [72].

Currently, the nanocapsules and nanospheres are the most common forms of polymer NPs, which are further divided into three categories: polymersomes, micelles, and dendrimers. Polymersomes are a type of artificial synthetic vesicles whose membranes consist of block copolymer amphiphiles [73]. These NPs contain an aqueous inner core surrounded by an outer bilayer membrane, which integrates hydrophobic drugs, while the core can encapsulate hydrophilic drugs, peptides, nucleotides, and enzymes. Moreover, the outer surface of the membrane can be modified to show the surface portion used for targeting [74]. They offer better stability and drug retention efficiency and become effective carriers for delivering therapeutic agents to cytosol [75,76]. Commonly used polymersomes include PEG [77], Poly (lactic-co-glycolic acid) (PLGA) [78], and poly (dimethylsiloxane) (PDMS) [79]. PEG polymers consist of repeated units of ethylene glycol, which can form linear or branched chain structures, with functional groups at one or more ends, enabling various conjugation possibilities and greatly increasing the drug loading [80]. Additionally, PLGA polymers are linear copolymers with repeating units of lactic and glycolic acid and the most widely applied type of particles due to their favorable properties, such as biocompatibility, biodegradability, and controllable drug release profile [81].

Polymer micelles can carry various types of therapeutics, such as small molecules or proteins, and have been widely used in clinical trials to deliver cancer drugs [82,83]. Dendrimers are a type of hyperbranched polymer with complex three-dimensional structures. Their mass, shape, size, and surface chemistry can be controlled in synthesis. Among them, polyethyleneimine (PEI) and poly (amidoamine) (PAMAM) dendrimers are widely used. Dendritic polymers can accommodate various types of therapeutics, and they are most commonly explored for transporting small molecules and nucleic acids [84]. In size, polymer micelles and other particles with a diameter of 10–100 nm are more likely to aggregate in tumors compared to larger liposomes [85]. Therefore, to achieve the most effective tumor permeability, it is crucial to control the particle size. However, polymeric NPs are still limited by the high risk of particle aggregation and toxicity. As a result, only a small amount of polymer nanomedicines has been approved by the Food and Drug Administration (FDA) for clinical use. Currently, polymeric NPs are being tested in a large number of clinical trials [60].

### 3.3. Inorganic NPs

Inorganic materials like gold, silica, manganese (Mn), and iron have been synthesized and are widely used in various types of delivery. The formulation of these inorganic NPs is highly precise and can be designed into various sizes, geometry shapes, and structures. Currently, the most extensively investigated are gold NPs (AuNPs), which have flexible forms in practical synthesis applications, including nanospheres, nanostars, nanorods, nanoshells, and nanocages [86]. Another common class of inorganic NPs materials is iron oxide, which possesses size-related superparamagnetic properties and has been successfully utilized for drug delivery [87]. Mesoporous silica (MSNs) and calcium phosphate (CaP) are also common inorganic NPs that show promise as emerging nanocarriers for delivering various molecules to various target sites [88]. Manganese dioxide (MnO_2_) NPs are also among the most stable and functional inorganic nanomaterials, they are widely used as carriers for nucleic acid, protein, and drug delivery [89]. Most inorganic NPs have good biocompatibility and stability, thereby addressing the limitations that organic materials cannot overcome. Nevertheless, their clinical application is restricted due to their low solubility and high toxicity [87,90].

### 3.4. Other NPs

In addition to the above nanomaterials, others have also been developed for drug delivery, such as cell membrane (CM)-camouflaged nanocarriers and metal–organic frameworks (MOFs), etc.

Recently, CM-camouflage technology has emerged as a new type of nanocarrier that provides NPs with the desired functions and complements the therapeutic efficacy [55,91,92]. For instance, tumor CM-decorated NPs carry abundant tumor antigens, which activate dendritic cells (DCs) and T cells to stimulate and infiltrate the tumor microenvironment, ultimately inhibiting tumor growth [92,93]. Additionally, a T lymphocyte membrane-decorated epigenetic nanoinducer loaded with IFN I inducer ORY-1001 and overexpressing PD-1 could identify and enter PD-L1-expressing cells. This would then provide intratumor IFN supplementation and inhibit its immunosuppressive activities, resulting in improved T cell-mediated antitumor activity [92]. Additionally, T-cell-membrane-coated NPs (TCMNPs) have been developed to target tumors and block immune checkpoint interactions. TCMNPs have shown potential as an alternative to current immunotherapy [94,95]. Moreover, red blood cell (RBC) membrane-coated NPs loaded with an anti-inflammatory Glyburide and monocyte membrane-coated NPs loaded with Gliclazide have also been developed for atherosclerosis therapy, respectively [96,97].

Cell-derived nanovesicles (CDNs) are artificially generated vesicles from the membranes of various immune cells. These vesicles reserve membrane proteins, resulting in low immune recognition [98,99,100]. Unlike extracellular vesicles (EVs), cell-derived nanovesicles overcome challenges such as low yields and can achieve higher yields using methods like mechanical extrusion, ultrasonic, or microfluidic [101,102,103]. Moreover, cell-derived nanovesicles can efficiently load RNA and modify surface proteins [104]. Since they originate from the cell membrane, these vesicles offer the possibility of producing vesicles expressing certain surface molecules [105,106]. Therefore, cell-derived nanovesicles provide a promising approach for enhancing immunomodulation through engineering.

Metal–organic frameworks (MOFs) are a highly versatile enzyme carrier [107,108]. They can encapsulate functional enzymes, potentially preserving their catalytic activity and protecting them from degradation by the surrounding environment [109]. In addition, MOFs offer advantages like a nontoxic or less toxic adjustable structure and pore size, large surface areas, and better biocompatibility, making them a potential delivery carrier for the development of nanoreactors [110,111,112,113].

## 4. Nanoparticle-Based Immunotherapy in Reversing T-Cell Exhaustion

Besides targeting immune checkpoints, NP-based remodeling of the tumor microenvironment and targeting the T cell metabolism have emerged as methods to reverse T cell exhaustion (Figure 1). The approaches that have been applied up to date are as follows (Table 1).

### 4.1. Targeting Checkpoint Blockade

Immune checkpoint blockade therapy has shown great promise for overcoming exhaustion. During T-cell activation, PD-1 is induced later and, after binding to PD-L1 or PD-L2, weakens TCR signaling through the recruitment of tyrosine phosphatase [114]. Anti-PD-1 mainly induces the expansion of specific tumor-infiltrating exhaust-like CD8 T cell subsets. Additionally, CTLA-4 is immediately upregulated after competitively binding to the B7 ligand, thereby limiting T-cell activation [115]. Anti-CTLA4 promotes the expansion of ICOS^+^ Th1-like CD4 effector cell populations [116]. Considering the constitutive expression of CLTA-4 on CD4 regulatory T cells (Treg), anti-CTLA4 mainly depletes inhibitory Treg cells [117]. However, there is still limited efficacy, significant toxicity, limited delivery potential, off-target effects, etc. [118]. NPs can greatly enhance delivery by protecting immunotherapies and enhancing the interaction with immune cells, ultimately enhancing the effectiveness of current immunotherapy approaches [119]. By modifying antibodies and other ligands on the surface of NPs, the specific and efficient uptake of NPs can be induced [120].

Inhibitory receptors are often targeted for NP-based immunotherapy against exhausted T cells. Recently, researchers have successfully transported PD-1 siRNA into T lymphocytes by lipid-coated CaP, improving the cellular uptake of siRNA and reducing PD-1 expression [121]. In addition, the direct delivery of PD-1 siRNA into T cells through AuNPs coated with PAMAM dendrimers could increase PD-1 gene silence and regulate T cell exhaustion. Considering that 2,3-dioxygenase (IDO), an immunosuppressive agent, causes exhaustion and the increased formation of regulatory T cells (Tregs), synergizing with an IDO inhibitor can further improve the tumor immunotherapeutic potency and reverse T cell exhaustion [49]. Meanwhile, some studies have also developed tumor CM-camouflaged NPs, which are edited by a long noncoding RNA (lncRNA). They way in which they synergistically act with anti-TIM-3 could amplify DC inflammasome activation by enhancing antigen cross-presentation and ameliorate exhaustion, showing remarkable efficacy against tumors [122].

Ionizable LNPs have been developed to deliver Epstein–Barr virus (EBV) latent membrane protein 2 (LMP2) mRNA to lymph nodes. Subsequently, LMP2 mRNA is expressed on antigen-presenting cells (APC), which activate CD8^+^ T cells, combat LMP2-expressing cancer cells, and promote the formation of memory T cells. Additionally, synergistic anti-PD-1 therapy could block the PD-L1 pathway, provoke strong anti-tumor efficacy, and reverse T-cell exhaustion [123]. Furthermore, a nanoparticle vaccine based on CaP, loaded with CpG, which is the Toll-like receptor 9 (TLR9) ligand, and a Gag epitope from a Friend retrovirus-specific CD8^+^ T cell, is very effective in activating DC and enhancing cell response [124,125,126]. Additionally, combining anti-PD-L1 with a therapeutic vaccination is more effective in reactivating the CD8^+^ T cell response and eliminating the viral [127].

T-cell-derived nanovesicles also provide an effective strategy to influence T-cell exhaustion. These nanovesicles generated by cytotoxic T cells are continuously extruded through membranes containing micro/nano pores. The surfaces of this nanovesicle are equipped with PD-1 and TGF-β receptors, which can cut off the PD-L1 pathway on cancer cells and clear TGF-β secretion, ultimately killing cancer cells and preventing cytotoxic-T-cell exhaustion [128]. To address the limited therapeutic effectiveness of immune checkpoint blockades, researchers have developed T-cell-membrane-coated NPs (TCMNPs). These TCMNPs contain proteins derived from the T-cell membrane and are modified with adhesion proteins LFA, allowing them to target tumors and block immune checkpoint interactions [94]. Additionally, they are loaded with anticancer drugs, such as dacarbazine, which can be released to kill cancer cells and induce FasL-mediated apoptosis, similar to the way cytotoxic T lymphocytes (CTLs) function. However, TCMNPs do not respond to immunosuppressive molecules like TGF-β1 and PD-L1, as they are capable of clearing them [95].

Besides antibody blockades, various alternatives to antibodies, such as PD-L1 aptamers and nanocarriers, are being developed to reduce the cost of tumor immunotherapy. Some studies have employed PD-L1 aptamer to modify gold nanorods (GNRs) to create a PD-L1-targeting therapy. This novel approach can block immune checkpoints, facilitate nanoparticle accumulation, and generate strong photoacoustic signals within tumors. When combined with concurrent photothermal therapy, this strategy significantly enhances antitumor immunity by activating CD8^+^ T cells and inhibiting Treg cells, thereby resulting in the suppression of exhaustion [129].

### 4.2. Remodeling the Tumor Microenvironment

The components within the tumor microenvironment, which participate in regulating the progress of T-cell exhaustion are garnering increasing attention as potential immune targets [41,130,131]. Tumor-associated macrophages (TAMs) are the most abundant types of immune cells in the tumor microenvironment [132,133]. TAMs, a vital component in this microenvironment, can influence tumor development [134]. TAMs are divided into M1 and M2 types. M1-like TAMs are typically considered tumor-killing macrophages, while the M2 type displays immunosuppressive properties, tumor-promoting functions, and distal metastasis [135]. Additionally, the tumor microenvironment also comprises various regulatory immunosuppressive cells, such as Tregs, myeloid-derived suppressor cells (MDSCs), and regulatory DCs. These cells play a role in tumor immune escape and pose a significant challenge in cancer immunotherapy [55]. Therefore, remodeling the tumor microenvironment by targeting TAMs, which includes inducing M2-to-M1 repolarization, inhibiting TAMs recruitment and depleting TAMs, and targeting immunosuppressive cells, has emerged as a strategy for cancer therapy and reversing exhaustion [136].

#### 4.2.1. Targeting TAMs


(1)Inducing TAMs repolarization


The majority of macrophages in the tumor microenvironment exhibit an anti-inflammatory, M2-like phenotype, and the number of M2-like TAMs is associated with poor prognosis and drug resistance [137,138]. Therefore, repolarizing M2-type TAMs to M1-type is beneficial for macrophages to exert tumor-killing effects, prevent tumor metastasis, and improve the immunosuppressive state of the tumor microenvironment. This TAMs reprogramming strategy might be a superior therapeutic approach.

Modulating the tumor microenvironment by consuming lactate and amplifying immunogenic cell death-induced immune responses can enhance the anti-tumor activity of cytotoxic T cells [139]. Lactic acid secreted by cancer cells promotes macrophage polarization (from the M1 to M2 phenotype) and T-cell exhaustion [140]. Wang H, et al., use MOFs to load lactate oxidase and oxaliplatin, which catalytically consume lactic acid and induce immunogenic cell death, respectively, and then coat with the platelet membrane (PM) for targeting tumor sites. This induces M2-to-M1 repolarization and decreases the Treg levels, thereby favoring tumor eradication [139].

Remodeling the tumor microenvironment combined with immune checkpoint blockades provokes a strong anti-tumor effect by reversing T-cell exhaustion [141]. Some studies utilize a sheddable PEG-decorated nanodrug loaded with a STAT6 inhibitor (AS) to induce M2-to-M1 repolarization. This contributes to recruiting effector T cells for tumor infiltration. Additionally, being combined with Galectin-9 blocker enhances the immune response and reduces exhaustion in highly malignant breast cancer [46]. In addition, the delivery of TLR7/8 agonist (R848)-loaded β-cyclodextrin NPs to TAMs in vivo has been demonstrated in multiple tumor models to promote M2-to-M1 repolarization, leading to the inhibition of tumor growth. Similarly, when used in combination with anti-PD-1 therapy, immunotherapy becomes more effective [142]. Moreover, the delivery of another TLR7/8 agonist 3M-052 to TAMs also induces phenotypic changes in TAMs and promotes tumor regression [143].

Targeting the stimulators of interferon genes (STING) pathway in TAMs through optimizing the delivery system is critical for reversing T-cell exhaustion. Cyclic dimeric guanosine monophosphate (c-di-GMP), a STING agonist, initiates a type I interferon (IFN-I) response, which causes CD8^+^ T cells to accumulate and infiltrate at the tumor sites and triggers an immunogenic response [144,145]. Utilizing novel NPs consisting of neutral cytidinyl lipid DNCA together with cationic lipid CLD to loaded c-di-GMP could stimulate more IFN-β production and promote immunogenic cell death, effectively reversing T-cell exhaustion in tumors [146].

Additionally, altering the phenotype of TAMs can also be achieved by targeting signaling pathways involved in macrophage polarization, such as histone deacetylases (HDACs), phosphoinositide 3-kinase gamma (PI3Kγ) inhibitors, etc. [136]. The high expression of HDAC has been observed in various types of cancers [147]. Recently, the HDAC inhibitor TMP195 has been found to change the TAMs phenotype, leading to a decrease in tumor burden and metastases. Furthermore, the combination of TMP195 and anti-PD-1 has exhibited additive effects on anti-tumor effects [148]. Moreover, PI3Kγ acts to enhance immunosuppressive activity and reduce immunostimulatory activity during inflammation and cancer. PI3Kγ inhibitors (SF1126 and AZD3458) or the genetic deletion of *Pik3cg*, can also facilitate macrophage reprogramming [149,150]. When AZD3458 is combined with anti-PD-1/anti-PD-L1, it exhibits greater therapeutic efficacy compared to the checkpoint inhibitor alone [136].


(2)Inhibiting TAMs recruitment


Chemokines play a crucial role in regulating the recruitment of TAMs to the inflammation site. Numerous studies have demonstrated that the chemokine ligands secreted by both cancer and stromal cells in the tumor microenvironment are significant in recruiting TAMs [151,152]. Consequently, blocking chemokine signals could be a novel strategy to disrupt the accumulation of TAMs and enhance therapeutic responses.

TAMs-recruiting chemokine (CCL2, CCL3, CCL4, and CCL5), VEGF, and CSF-1 have the potential to enhance TAMs recruitment and serve as therapeutic targets. Among them, CCL2 and its ligand CCR2 (the CCL2–CCR2 axis) are crucial in determining TAMs accumulation [153]. To block this axis, cationic NPs encapsulating siRNA-CCR2 (CNP/siCCR2) have been developed to inhibit CCR2 expression in monocytes. More importantly, by blocking the recruitment of monocytes to tumor tissue, CNP/siCCR2 can reprogram the tumor microenvironment, suppress tumor growth, reduce tumor metastasis, and exert effective anti-tumor effects [154]. Furthermore, CCR2 antagonists (RS504393 and RS102896) have been developed to suppress tumor metastasis [155,156]. In some mouse tumor and metastasis models, CCR2 antagonists synergize with anti-PD-1 therapy to show an enhanced antitumor response [157].


(3)TAMs depletion


The infiltration of TAMs in tumor tissue is significantly negatively correlated with tumor prognosis. TAMs generate signals that support tumor growth and promote cell survival. When TAMs are depleted, the production of these signals is reduced, leading to a decrease in tumor cell proliferation [158]. Therefore, the targeted depletion of M2-like TAMs is a feasible option for immunotherapy. Some studies demonstrate that TAMs induce exhaustion programs while depleting TAMs reduces exhaustion programs and enhance the effectiveness of tumor-infiltrating CD8^+^ T cells [159]. TAMs are specifically targeted for apoptosis induced by liposomal clodronate due to their active phagocytosis of liposomes [160]. Using liposomes encapsulating clodronate could deplete TAMs, thereby restoring the host’s defenses and eliminating tumor cells [158,160]. The targeted delivery of proapoptotic peptides M2pep to TAMs to selectively reduce the TAMs population has shown improved survival rates and anticancer effects [161].

In addition, the production and activation of TAMs mainly rely on macrophage colony-stimulating factor-1 (CSF-1). Besides clodronate liposome treatment, CSF-1 receptor (CSF-1R) inhibitors BLZ945 and PLX3397 have also been shown to deplete macrophages. This depletion restores T-cell migration and infiltration into tumor islets and improves anti–PD-1 immunotherapy [162,163]. Furthermore, TAMs dual-targeting lipid NPs (M2NPs) loaded with CSF-1R siRNA have also been developed. These NPs have a scavenger receptor targeting the peptide connected to TAMs-targeting peptides (M2pep) on the surface. These dual-targeting NPs significantly reduce the proportion of TAMs, inhibit tumor growth, downregulate the expression of PD-1 and TIM-3, and restore T-cell function [164]. Compared to inhibiting TAMs recruitment, depleting pulmonary TAMs may be a favorable strategy for alleviating lung cancer progression [158]. However, further exploration is still needed.

#### 4.2.2. Targeting Immunosuppressive Cells

Although various engineered NPs have been designed to target effector cells, such as T cells and TAMs, other approaches have also been used to suppress the activity of immunosuppressive cells, such as Treg cells, MDSCs, and regulatory DCs, and indirectly enhance immune responses. For instance, hybrid NPs conjugated with the tLyp1 peptide and loaded with the tyrosine kinase inhibitor imatinib have been used to target the Nrp1 receptor on Treg cells and downregulate their suppression by inhibiting STAT3/STAT5 signaling. When these NPs are combined with anti-CTLA-4 therapy, they show enhanced tumor inhibition [165]. Furthermore, a lipid nanoparticle encapsulating the cyclin-dependent kinase inhibitor dinaciclib and modified with anti-PD-L1 has been designed to deplete MDSCs and attenuate their immunosuppressive functions [166]. When exhaustion occurs, TIM-3 is also expressed on DCs, which inhibits their response upon interaction with their ligands [167].

### 4.3. Targeting T-Cell Metabolism

NP-based targeting the T-cell metabolism has become a new method for reversing exhaustion. Impaired mitochondrial function and hypoxia are the main metabolic features and drivers of the exhaustion of T cells, along with the low expression of major histocompatibility complex class I (MHC I) on the surface of tumor cells. This deficiency causes the low-efficiency recognition of T cells, which compromises therapeutic outcomes. Zhang D, et al., utilize a tumor CM-decorated vesicle, modified by oxidized sodium alginate and loaded with axitinib, 4-1BB antibody, and PCSK9 inhibitor PF-06446846. Axitinib can alleviate hypoxia, the 4-1BB antibody can enhance T-cell mitochondrial biogenesis, and PF-06446846 increases the expression of MHC I and further enhances the efficiency recognition of T cells. The synergistic effects of these agents significantly revitalize T cell function [168].

Mitochondrial dysfunction is an intrinsic trigger factor for exhaustion. Wu H, et al. demonstrate that mitochondrial dysfunction drives T cells towards terminal exhaustion through maintaining stable levels of hypoxia-inducible factor 1α (HIF-1α) protein expression and the related glycolytic reprogramming [169]. Additionally, HIF-1α also initiates downstream gene PD-L1 transcription [170]. Since hypoxia and ROS are the main drivers in immune exhaustion, ROS-responsive manganese dioxide (MnO_2_) NPs are developed to carry the HIF-1α inhibitor (acriflavine) to the tumor sites, successfully relieving T-cell exhaustion and activating tumor-specific immune responses [171].

Overall, these data suggest that NP-based methods may have the potential to reverse T-cell exhaustion, but further investigation is still required.

**Table 1 ijms-25-01396-t001:** Nanoparticle-based immunotherapy in reversing T-cell exhaustion.

Strategy	Composition of NPs	Immunomodulators	Target Cells	Intervention Mechanism	Ref.
Targeting checkpoint blockade	PAMAM dendrimer-entrapped AuNPs	PD-1 siRNA; IDO inhibitor	T cells	Silencing PD-1 gene	[49]
	lncRNA-edited tumor CM-camouflaged NPs	Anti-TIM-3	DC cells and CD8^+^ T cells	Mediating antigen cross-presentation and dampening immunosuppression	[122]
	LMP2-mRNA LNPs	Anti-PD-1	CD8^+^ T cells	Enhancing memory T-cell formation	[123]
	CaP-based nanoparticle vaccine	Anti-PD-L1; CpG; a virus-specific CD8^+^ T cell epitope	CD8^+^ T cell	Reactivating CD8^+^ T cell immunity	[127]
	T-cell-derived nanovesicles	PD-1; TGF-β receptor	Cancer cells	Blocking the PD-L1 pathway and eliminating TGF-β	[128]
	T-cell-membrane-coated NPs (TCMNPs)	LFA; Dacarbazine	Tumor cells	Blocking immune checkpoint interactions and inducing FasL-mediated apoptosis	[95]
	Gold nanorods (GNRs)	PD-L1 aptamer	Tumor cells	Activating CTLs and inhibiting Treg cells	[129]
Remodeling the tumor microenvironment	MOFs coating with PM	Lactate oxidase; Oxaliplatin	TAMs	Promoting M2-to-M1 repolarization and decreasing Treg levels	[139]
	PEG-decorated NPs	Anti-Galectin-9; AS	TAMs	Promoting M2-to-M1 repolarization and enhancing effector T-cell infiltration	[46]
	Cyclodextrin NPs	TLR7/8 agonist (R848); anti-PD-1	TAMs	Promoting M2-to-M1 repolarization and inhibiting tumor growth	[142]
	Neutral cytidinyl lipid DNCA/cationic lipid CLD	c-di-GMP	Cancer cells	Triggering immunogenic cell death and increasing effector T-cell infiltration	[146]
	Cationic NPs	CCR2 siRNA	Monocytes	Inhibition of TAMs recruitment	[154]
	Liposome	Clodronate	TAMs	TAMs depletion	[160]
	Lipid NPs	CSF-1R siRNA; M2pep; a scavenger receptor targeting peptide	TAMs; scavenger receptor	TAMs depletion	[164]
	Hybrid NPs	tLyp1 peptide; Imatinib; Anti-CTLA-4	Tregs	Downregulating Tregs suppression	[165]
	Lipid NPs	Dinaciclib; Anti-PD-L1	MDSCs	Depleting MDSCs and attenuating their immunosuppressive functions	[166]
Targeting T-cell metabolism	Tumor CM decorated vesicle	Axitinib; 4-1BB antibody; PF-06446846	T cells	Promoting T-cell mitochondrial biogenesis and reducing hypoxia	[168]
	MnO_2_ NPs	Acriflavine	Tumor cells	HIF-1α functional inhibition and subsequently activating tumor-specific immune responses	[171]

## 5. Conclusions and Perspectives

T-cell exhaustion commonly emerges in numerous pathogens, infections, and cancer. T-cell exhaustion usually leads to disease progression. Although multiple mechanisms may be involved in the occurrence of exhaustion, it is still necessary to dissect how these mechanisms network together to influence the immune response and explore new targets for immunotherapy.

An NP-based programming approach has been studied in the process of rejuvenating T-cell exhaustion. The results demonstrate that the NP-based combination immunotherapy elicits strong T-cell responses and reverses T-cell exhaustion. Despite these investigations enhancing our understanding of the mechanisms of exhaustion and introducing new research on NP-targeted therapies, numerous important questions remain unanswered. For instance, while targeting pathways like PD-1 antibodies have shown some therapeutic effects, we still know very little about the underlying mechanisms. Additionally, when targeting multiple pathways to reverse T-cell exhaustion, we lack a comprehensive molecular understanding of their synergistic effects.

In addition, there still exist several limitations in NPs’ application in the delivery system. For instance, toxicity, low intake, off-targeted, tissue retention-induced immune tolerance, etc., [172]. Cytotoxicity is the most common one [173]. NPs also exhibit immunogenicity and can be easily recognized and cleared by immune cells [174]. Furthermore, the size of NPs also affects cell uptake, as it influences the enthalpy and entropy capabilities that control the adsorption effect of NPs [175]. Therefore, certain-sized non-degradable NPs could be retained in tissue and organs, such as lungs, liver, kidneys, etc., and pose serious hazards [65].

Rationale NP designs are critical for improving precision therapies. This review has discussed numerous NP designs for reversing T-cell exhaustion. The NPs platform offers a range of modifiable characteristics, such as size, shape, surface properties, charge, and responsiveness, which can be selected to optimize specific applications in chronic infection and cancer treatment. For instance, surface modifications are implemented in some NP designs to prevent the side-effect of non-specific distribution. Additionally, many NPs incorporate PEG to avoid rapid excretion. However, the most important issue remains to be that by understanding the characteristics of exhausted T cells and their immunosuppressive microenvironment during the exhaustion process, NPs can be designed for targeted interventions to achieve the best outcomes.

Notably, the NP-based approach has shown impressive and remarkable outcomes during preclinical research, indicating its strong potential for combating cancer and infectious diseases. However, only a few materials have been examined in clinical trials so far, and none have been authorized for use [176]. Additionally, NP-based antigen delivery can induce immune tolerance, which promotes the application of NPs in autoimmunity [177,178]. Therefore, further investigation into the pathogenesis of exhaustion and NP-based immunotherapies is necessary for developing novel interventions against exhaustion.

## Figures and Tables

**Figure 1 ijms-25-01396-f001:**
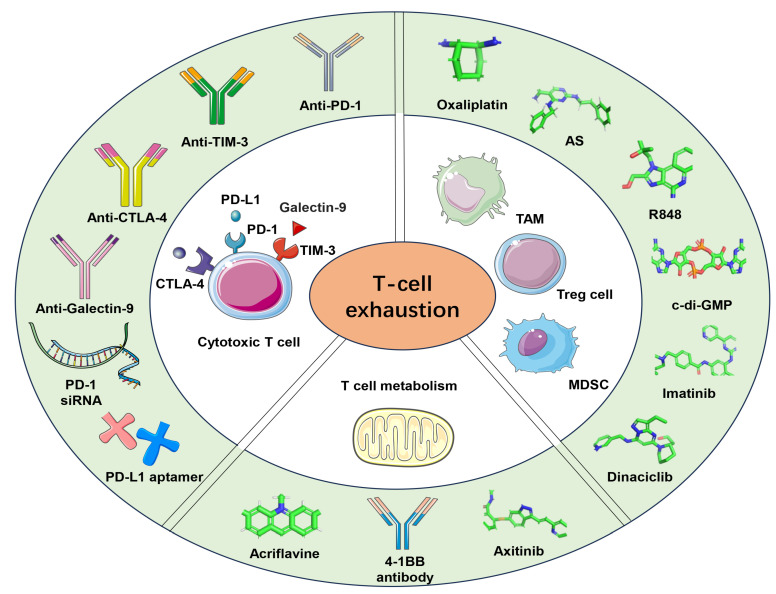
NP-based immune-targeting methods to reverse T cell exhaustion. These include targeting inhibitory checkpoint blockades, remodeling the tumor microenvironment, and targeting the metabolism of exhausted T cells, etc. NPs are used to deliver antibodies like anti-PD-1, anti-TIM-3, anti-Galectin-9, PD-1 siRNA, PD-L1 aptamer, etc., which can block immune checkpoints. Additionally, c-di-GMP and small-molecule-targeting therapeutics like oxaliplatin, AS, TLR7/8 agonist (R848), CCR2 siRNA, and clodronate can induce M2-to-M1 macrophage repolarization and trigger efficient immunity. NP-loaded imatinib and dinaciclib are designed to target immunosuppressive cells, such as Tregs and MDSCs. Acriflavine, axitinib, or 4-1BB antibody can alter T cell metabolism, promoting T cell mitochondrial biogenesis or reducing hypoxia, thus effectively relieving T-cell exhaustion. Furthermore, NP-based combination therapies elicit strong antitumor responses.

## Data Availability

Data sharing not applicable.

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
