# Peer review of "Nanoparticle-Based Immunotherapy for Reversing T-Cell Exhaustion"

_ijms, 2024, doi:10.3390/ijms25031396_

Round 1

Reviewer 1 Report

Comments and Suggestions for Authors

This manuscript summarized the latest insight into the mechanisms involved in T cell exhaustion, and brief discussed nanoparticle-based immunotherapies that aim to reverse T cell exhaustion. Although interesting, some suggestions are provided for the author’s consideration:

1.     Based on the content, I would suggest changing the title to “Nanoparticle-based Immunotherapy for Reversing T-cell Exhaustion”

2.     Since antibodies that blocking immune checkpoints, such as PD-1, PD-L1, or CLTA-4, are the most common strategy, I would suggest the authors to focus on how NPs assist the delivery of antibodies as well as the design rationale.

3.     Immunosuppressive tumor microenvironment is a complicated collection consisting of neoplastic cells and non-neoplastic cells, why the authors focus on tumor-associated macrophages (TAMs) particularly? How about other cells?

4.     Also, the summary of TAMs-targeting therapies is insufficient, such as TAMs depletion, inhibition of TAMs recruitment as well as TAM re-polarization, which is limited to AS or c-di-GMP.

5.     By summarizing these nanoparticle-based immunotherapies, what is the rationale to design nanoparticles or how to improve their delivery efficiency? Additional discussion is required.

6.     Some related works are encouraged to be cited: Exploration, 2022, 2:20210157; Chin. Chem. Lett. 2023, 34, 107518.

Author Response

Dear Editor,

Thanks for your careful consideration of our manuscript. We appreciate the helpful comments from reviewers very much. We have made careful revisions accordingly. Our point-by-point responses are as follows.

Reviewer 1

This manuscript summarized the latest insight into the mechanisms involved in T cell exhaustion, and brief discussed nanoparticle-based immunotherapies that aim to reverse T cell exhaustion. Although interesting, some suggestions are provided for the author’s consideration:

  1. Based on the content, I would suggest changing the title to “Nanoparticle-based Immunotherapy for Reversing T-cell Exhaustion”

Answer: Thanks for your great suggestion. The title may be not accurate, we have corrected it to “Nanoparticle-based Immunotherapy for Reversing T-cell Exhaustion”.

  1. Since antibodies that blocking immune checkpoints, such as PD-1, PD-L1, or CLTA-4, are the most common strategy, I would suggest the authors to focus on how NPs assist the delivery of antibodies as well as the design rationale.

Answer: Thanks for your constructive comments. Although blocking immune checkpoints, such as PD-1-targeted therapies, are now licensed to treat human cancers, not all patients respond equally to immune checkpoint therapies. Furthermore, there are still challenges with efficacy, patient variability, and off-target effects. Therefore, how to improve PD-1-targeted therapies during chronic M. tuberculosis infection is urgently needed.

Nanoparticles (NPs) are capable of combining immune checkpoint blockade with other immunotherapies to enhance T cell immunity and control pathogens, overcoming T cell dysfunction and immunosuppression. NPs also have the potential to significantly improve delivery by protecting immunotherapeutics and enhancing their interaction with immune cells. The combination therapy synergistically reactivates T cell immunity compared to immune checkpoint blockade alone. Therefore, the rational NP design can improve cargo delivery or remodel microenvironments and thus increase the efficacy of existing therapies. As you suggested, we have added some words about NP designs when elaborating the strategy.

  1. Immunosuppressive tumor microenvironment is a complicated collection consisting of neoplastic cells and non-neoplastic cells, why the authors focus on tumor-associated macrophages (TAMs) particularly? How about other cells?

Answer: Thanks for your comments. Yes, you’re right. The tumor microenvironment is comprised of a variety of cells, including neoplastic cells and non-neoplastic cells. Tumor-associated macrophages (TAMs) are the most abundant types of immune cells in the tumor microenvironment. Moreover, TAMs are key components in the microenvironment and play a crucial role in tumor-promoting inflammation and progression. Additionally, the tumor microenvironment also comprises several regulatory immunosuppressive cells, such as Tregs, myeloid-derived suppressor cells (MDSCs), and regulatory DCs, which contribute to tumor immune escape and are the major challenge in cancer immunotherapy. Therefore, remodeling the tumor microenvironment through targeting TAMs, which includes inducing M2-to-M1 repolarization, inhibiting TAMs recruitment, and depleting TAMs, and targeting immunosuppressive cells, has emerged as strategies for cancer therapy and reversing exhaustion. We have added several paragraphs about the possible strategies to 4.2. section to make this review more comprehensive.

  1. Also, the summary of TAMs-targeting therapies is insufficient, such as TAMs depletion, inhibition of TAMs recruitment as well as TAM re-polarization, which is limited to AS or c-di-GMP.

Answer: Thanks for your suggestion. Yes, the TAMs-targeting therapies are insufficient. We have added several paragraphs about TAMs-focused therapeutic strategies in the manuscript, to make it more clearly.

  1. By summarizing these nanoparticle-based immunotherapies, what is the rationale to design nanoparticles or how to improve their delivery efficiency? Additional discussion is required.

Answer: Thanks for your constructive comments. We should focus on how NPs assist in the delivery of antibodies as well as the design rationale. We have added several words about NP design to 4. Nanoparticle-based immunotherapy in reversing T-cell exhaustion section and discussion section to make it better.

  1. Some related works are encouraged to be cited: Exploration, 2022, 2:20210157; Chin. Chem. Lett. 2023, 34, 107518.

Answer: Thanks for your great suggestion. We carefully read the above literature and cite these studies in the manuscript to make this review more clearly.

We have revised our manuscript very carefully. If there are any questions, please let us know and we will do more revisions. We are looking forward to hearing from you.

Best wishes,

Fei Li, M. D.

Reviewer 2 Report

Comments and Suggestions for Authors

In this study, authors have reviewed the strategy for removing the dis function problem of T cell exhaustion using nanoparticle based immunotherapy.

In this regard, they mention the different types of nanoparticles and strategies for reversing T cell exhaustion.

However, authors should address the issues:

1-      In the abstract, the final purpose of the study should be mentioned, for example treatment of cancer or chronic diseases.

2-      As a checkpoint, authors must explain the anti CTLA4 therapeutics and its role in recovery of T cell exhaustion in the subtitle of “4.1. Targeting checkpoint blockade”

3-      Authors should mention the biomimetic strategy have been used in nanoparticles section. In this regard they can cite the below articles:

https://pubmed.ncbi.nlm.nih.gov/32812291/

https://www.nature.com/articles/s41598-023-41136-y

https://www.sciencedirect.com/science/article/pii/S2666542523000681

and for design of PLGA nanoparticles:

https://www.hindawi.com/journals/jnm/2022/6165759/

4-      Please change the subtitle of “Nanoparticle is widely used”

5-      More comparisons with the recent articles are needed. Conclusion and finding of the review must be strongly presented. The manuscript should be expanded more.

Best regards.

Comments on the Quality of English Language

Moderate editing of English language required

Author Response

Dear Editor,

Thanks for your careful consideration of our manuscript. We appreciate the helpful comments from reviewers very much. We have made careful revisions accordingly. Our point-by-point responses are as follows.

Reviewer 2

In this study, authors have reviewed the strategy for removing the dis function problem of T cell exhaustion using nanoparticle based immunotherapy.

In this regard, they mention the different types of nanoparticles and strategies for reversing T cell exhaustion.

However, authors should address the issues:

1-In the abstract, the final purpose of the study should be mentioned, for example treatment of cancer or chronic diseases.

Answer: Thanks for your suggestion. We have added the final purpose (see below) of the manuscript to the abstract section to make it better.

This data could aid in comprehending the immunopathogenesis of exhaustion and accomplishing the objective of preventing and treating chronic diseases or cancer.

2- As a checkpoint, authors must explain the anti CTLA4 therapeutics and its role in recovery of T cell exhaustion in the subtitle of “4.1. Targeting checkpoint blockade”

Answer: Thanks for your suggestion. As a checkpoint molecule, CTLA-4 inhibits the positive co-stimulatory signaling of CD28 by competitively binding to CD80 and CD86, resulting in inhibiting T cell responses. Moreover, CTLA-4 binds to CD80/CD86 with an affinity 20-fold higher than CD28. Besides that, several studies showed constitutive expression of CLTA-4 on CD4 regulatory T cells (Treg). Anti-CTLA4 predominantly inhibits Treg cells, thereby increasing the CD8 T-cell to Treg (CD8/Treg) ratio. Anti-CTLA4 promotes the expansion of T cells and increases cytokine production. In addition, some studies demonstrate that anti-CTLA-4 induces the expansion of Th1-like CD4 effector population in addition to engaging specific subsets of exhausted-like CD8 T cells.

We have added these CTLA-4 inhibitory mechanisms, anti-CTLA4 therapeutics, and their role in recovery of T cell exhaustion to the 4.1. section.

References:

[1] Apol ÁD, Winckelmann AA, Duus RB, Bukh J, Weis N. The role of CTLA-4 in T cell exhaustion in chronic hepatitis B virus infection. Viruses. 2023 May 10;15(5):1141.

[2] Twyman-Saint Victor C, Rech AJ, Maity A, Rengan R, Pauken KE, Stelekati E, Benci JL, Xu B, Dada H, Odorizzi PM, Herati RS, Mansfield KD, Patsch D, Amaravadi RK, Schuchter LM, Ishwaran H, Mick R, Pryma DA, Xu X, Feldman MD, Gangadhar TC, Hahn SM, Wherry EJ, Vonderheide RH, Minn AJ. Radiation and dual checkpoint blockade activate non-redundant immune mechanisms in cancer. Nature. 2015 Apr 16;520(7547):373-7.

[3] Wei SC, Levine JH, Cogdill AP, Zhao Y, Anang NAS, Andrews MC, Sharma P, Wang J, Wargo JA, Pe'er D, Allison JP. Distinct cellular mechanisms underlie anti-CTLA-4 and anti-PD-1 checkpoint blockade. Cell. 2017 Sep 7;170(6):1120-1133.e17.

3- Authors should mention the biomimetic strategy have been used in nanoparticles section. In this regard they can cite the below articles:

https://pubmed.ncbi.nlm.nih.gov/32812291/

https://www.nature.com/articles/s41598-023-41136-y

https://www.sciencedirect.com/science/article/pii/S2666542523000681

and for design of PLGA nanoparticles:

https://www.hindawi.com/journals/jnm/2022/6165759/

Answer: Thanks for your great comments. We carefully read the above literature. Then we updated and added these references to the manuscript.

4- Please change the subtitle of “Nanoparticle is widely used”

Answer: Thanks for your suggestion. I have changed the subtitle to “Nanoparticle classification and application” to make it more clearly.

5- More comparisons with the recent articles are needed. Conclusion and finding of the review must be strongly presented. The manuscript should be expanded more.

Answer: Thanks for your suggestions. We have added more recent articles, corrected the conclusion of the manuscript, and further expanded the manuscript.

We have revised our manuscript very carefully. If there are any questions, please let us know and we will do more revisions. We are looking forward to hearing from you.

Best wishes,

Fei Li, M. D.

Reviewer 3 Report

Comments and Suggestions for Authors

In their manuscript entitled "Nanoparticle-based Immunotherapy in T-cell Exhaustion" the authors in detail nanoparticle-based immunotherapies that aim to reverse T cell exhaustion.

The manuscript is well written and discusses major points.

However, to impove the impact and scientific value of this highly interesting topic I recommend the following suggestions:

- The indroduction leading the the topic could be mire detailed. So, for better ubderstanding fpr the scientific readership, not very familiar to this topic, but also ver interested,  I suggest to get more in detail with the refences 1-5 mentionend.

- Not only the organic NP´s have some signifinant toxic side effects in the body (as the aiuthors correctly mention) , als orgnanic NPs (lipid based and polymeric NPs) have unwanted or "unpredictible" side effects. So, the authos shouls also provide the limitation in the application (according th the current state of the art) in living systems.

- Interesting would be an additional graphical sketch to visualize the "mode" of binding of the NP to the T-Cell.

- 4.1. Targeting checkpoint blockade  and  4.2. Remodeling the tumor microenvironment: here,  also a sketch of an selected exa,ple would contribute to a better understanding.

In general, the manuscript discusses an very interesting topic in medicine of an unmet medical need. I recommend for publication after discussing the mentioned points.

Comments on the Quality of English Language

minor corrections

Author Response

Dear Editor,

Thanks for your careful consideration of our manuscript. We appreciate the helpful comments from reviewers very much. We have made careful revisions accordingly. Our point-by-point responses are as follows.

Reviewer 3

In their manuscript entitled "Nanoparticle-based Immunotherapy in T-cell Exhaustion" the authors in detail nanoparticle-based immunotherapies that aim to reverse T cell exhaustion.

The manuscript is well written and discusses major points.

However, to impove the impact and scientific value of this highly interesting topic I recommend the following suggestions:

- The indroduction leading the the topic could be mire detailed. So, for better ubderstanding fpr the scientific readership, not very familiar to this topic, but also ver interested,  I suggest to get more in detail with the refences 1-5 mentionend.

Answer: Thanks for your great suggestion. We carefully read the references 1-5 mentioned carefully, and further added paragraphs (see below) to expand the introduction section to make it easier to understand.

During acute infections, naive T cells are activated and undergo differentiation into effector T cells and memory T cells. Effector T cells play a role in eliminating antigens and controlling infections. Following antigen clearance, the majority of effector cells would die by apoptosis. Only about 5%-10% of T cells persist and continue to differentiate into memory T cell subsets. These memory T cells have long-term survival and retain the capability of homeostatic proliferation. when re-stimulation, memory T cells can generate effector T cells and sustain a recall response.

By contrast, during several chronic infections and cancer, due to persistent antigen stimulation, antigen-specific T cells become dysfunctional or even exhausted, which is characterized by decreased effector function, sustained expression of various inhibitory receptors, such as PD-1, TIM-3, and cytotoxic T lymphocyte antigen 4 (CTLA-4), and loss of memory ability. T-cell exhaustion usually leads to disease progression. Recently, targeted immune checkpoint molecules have been widely studied and can effectively reverse T-cell exhaustion. Although immune checkpoint blockade therapy shows great promise, there are still limitations, such as limited response rates, the possibility of relapse, and toxicity. Therefore, it is necessary to summarize the characteristics of exhaustion to explore new therapeutic targets for re-versing T-cell exhaustion in chronic infections and cancer.

- Not only the organic NP´s have some signifinant toxic side effects in the body (as the aiuthors correctly mention) , als orgnanic NPs (lipid based and polymeric NPs) have unwanted or "unpredictible" side effects. So, the authos shouls also provide the limitation in the application (according th the current state of the art) in living systems.

Answer: Thanks for your suggestion. Yes, the advantages of organic NPs cannot be ignored. We have added the limitation of lipid-based and polymeric NPs to the manuscript.

- Interesting would be an additional graphical sketch to visualize the "mode" of binding of the NP to the T-Cell.

Answer: Thanks for your suggestions. Yes, it is more clear to use a graphical sketch to describe the "mode" of NP binding to T-cells. However, the model varies in NP type, surface modification, drug loading, and diseased microenvironments. Therefore, no separate figure was created, but massive explanations of NP design in the text and the content in the table were added to the manuscript.

- 4.1. Targeting checkpoint blockade  and  4.2. Remodeling the tumor microenvironment: here,  also a sketch of an selected exa,ple would contribute to a better understanding.

Answer: Thanks for your suggestion. We have added some examples to the 4.1 and 4.2 sections and extended the content of Figure 1 to make it more understandable.

In general, the manuscript discusses an very interesting topic in medicine of an unmet medical need. I recommend for publication after discussing the mentioned points.

Answer: Thanks for your evaluation.

We have revised our manuscript very carefully. If there are any questions, please let us know and we will do more revisions. We are looking forward to hearing from you.

Best wishes,

Fei Li, M. D.

Reviewer 4 Report

Comments and Suggestions for Authors

The review of Li and coworkers on nanoparticles-based immunotherapy in T-cell exhaustion promised more than finally exceeded. The current version needs a deep revision before its appearance.

The principal weakness of the manuscript is the unclear structure and missing information. The minor weaknesses also reduce the manuscript's value.

- It is funny that the authors repeat the title in the Keywords section. The majority of journals require this section for a reason. Please correct this serious problem.

- The restructuration of the manuscript is necessary. The function of the Introduction is to outline the purpose and importance of the manuscript. The current chapter is short, and as far as the referee could evaluate, chapter 2 (lines 46-108) has a better place in that section, either in a reduced length.

- The chapter 3 title seems strange, as a title rarely like that.

- The discussion of the NPs fights with serious challenges. It is odd in a review article that the authors do not identify the materials discussed. Chapter 3 is not more than a badly-written Wikipedia article. Some points have a better place in the Introduction.

What is the typical dimension of NPs used in immunotherapy?

What kind of lipids are in NPs?

Which polymers - chemical types and repeating units - are advantageous for NP construction?

What is the role of the NPs in immunotherapy? Which properties do the NPs make suitable NPs for immunotherapy?

- As the referee sees, the review section starts with chapter 4. The referee recommends to combine chapters 3 and 4.

Please clarify the meaning of 'remodeled' as its meaning is unclear. Why is M2-TAM transformation to M1-TAM remodeling? What is the transformation? Please explain it in detail.

Some NPs in Table 1 are not characterized (in column 2).

- The last section failed to mention the perspectives, trends, and future tasks.

- The references are incorrectly formatted and contain some inconsistencies. There are some abbreviated journal names. Abbreviated names need dots, and the full journal names are also incorrectly spelled.

Comments on the Quality of English Language

There are no major errors, but some polishing is needed, especially in the References section, where the multi-word journal titles are misspelled.

Author Response

Dear Editor,

Thanks for your careful consideration of our manuscript. We appreciate the helpful comments from reviewers very much. We have made careful revisions accordingly. Our point-by-point responses are as follows.

Reviewer 4

The review of Li and coworkers on nanoparticles-based immunotherapy in T-cell exhaustion promised more than finally exceeded. The current version needs a deep revision before its appearance.

The principal weakness of the manuscript is the unclear structure and missing information. The minor weaknesses also reduce the manuscript's value.

- It is funny that the authors repeat the title in the Keywords section. The majority of journals require this section for a reason. Please correct this serious problem.

Answer: Thanks for your suggestion. Yes, the keywords may be not accurate, we have corrected it to “T-cell exhaustion; nanoparticle; immune checkpoint blockade; tumor microenvironment; T cell metabolism”.

- The restructuration of the manuscript is necessary. The function of the Introduction is to outline the purpose and importance of the manuscript. The current chapter is short, and as far as the referee could evaluate, chapter 2 (lines 46-108) has a better place in that section, either in a reduced length.

Answer: Thanks for your constructive comments. The purpose and importance of the manuscript are to explore the immunopathogenesis of exhaustion and emphasize NP-based immunotherapy to accomplish the objective of preventing and treating chronic diseases or cancer. We have added some words (see below) to expand the introduction section to make it easier to understand.

During acute infections, naive T cells are activated and undergo differentiation into effector T cells and memory T cells. Effector T cells play a role in eliminating antigens and controlling infections. Following antigen clearance, the majority of effector cells would die by apoptosis. Only about 5%-10% of T cells persist and continue to differentiate into memory T cell subsets. These memory T cells have long-term survival and retain the capability of homeostatic proliferation. when re-stimulation, memory T cells can generate effector T cells and sustain a recall response.

By contrast, during several chronic infections and cancer, due to persistent antigen stimulation, antigen-specific T cells become dysfunctional or even exhausted, which is characterized by decreased effector function, sustained expression of various inhibitory receptors, such as PD-1, TIM-3, and cytotoxic T lymphocyte antigen 4 (CTLA-4), and loss of memory ability. T-cell exhaustion usually leads to disease progression. Recently, targeted immune checkpoint molecules have been widely studied and can effectively reverse T-cell exhaustion. Although immune checkpoint blockade therapy shows great promise, there are still limitations, such as limited response rates, the possibility of relapse, and toxicity. Therefore, it is necessary to summarize the characteristics of exhaustion to explore new therapeutic targets for re-versing T-cell exhaustion in chronic infections and cancer.

- The chapter 3 title seems strange, as a title rarely like that.

Answer: Thanks for your suggestion. I have changed the subtitle to “Nanoparticle classification and application” to make it more clearly.

- The discussion of the NPs fights with serious challenges. It is odd in a review article that the authors do not identify the materials discussed. Chapter 3 is not more than a badly-written Wikipedia article. Some points have a better place in the Introduction.

What is the typical dimension of NPs used in immunotherapy?

What kind of lipids are in NPs?

Which polymers - chemical types and repeating units - are advantageous for NP construction?

What is the role of the NPs in immunotherapy? Which properties do the NPs make suitable NPs for immunotherapy?

Answer: Thanks for your great suggestions. In size, NPs with a diameter less than 10 nm have generally been shown to be rapidly eliminated by the kidneys, whereas NPs larger than 200 nm risk activating the complement system, if not otherwise engineered. Therefore, the typical dimension of NPs used in immunotherapy is usually between 10-200nm. For instance, polymer micelles and other particles with a diameter of 10-100nm are more likely to aggregate in tumors than larger liposomes. Therefore, to achieve the best tumor permeability effect, it is important to control the particle size.

Lipid-based NPs typically consist of a phospholipid, an ionizable lipid, cholesterol, and a PEGylated lipid. The most typical representatives of Lipid-based NPs s are liposomes and lipid NPs (LNPs). Recent developed LNPs are mainly composed of ionizable lipids, helper lipids, PEG-lipids, and cholesterol.

PEG polymers are composed of repeating units of ethylene glycol, which can be produced as linear or branched chains, with functional groups at one or more termini to enable a variety of conjugation possibilities and greatly increase drug loading. Additionally, PLGA polymers are linear copolymers with repeating units of lactic and glycolic acid and the most widely-applied type of particles due to their favorable properties, such as biocompatibility, biodegradability, and controllable drug release profile.

The role of the NPs in immunotherapy, and which properties do the NPs make suitable NPs for immunotherapy are as follows: Nanoparticle (NP)-based delivery approaches can reduce side effects and toxicity on non-target cells in immunotherapy, thereby significantly improving the effectiveness of immunotherapy. NPs have remarkable features, such as adjustable structures, strong biomolecular loading capacity, abundant surface modification, and controllable release molecules. NPs-based delivery systems provide extended circulation and active targeting. They can target solid tumors by targeting tumor cells, stimulating or reprogramming immune cells, remodeling the tumor microenvironment, and altering immune responses, thereby generating effective antitumor immunity. Moreover, NPs can be easily modified to bind to specific receptors or ligands, thereby enhancing compatibility and efficiency. As a result, NPs-based strategies have gained widespread attention in disease treatment.

We have added several paragraphs about the above content to the manuscript to make it more comprehensive.

- As the referee sees, the review section starts with chapter 4. The referee recommends to combine chapters 3 and 4.

Answer: Thank you for your suggestion. I set up Chapter 3 mainly to provide a detailed description of the classification and application of NPs, and then transition to Chapter 4 on the application of NPs in T cell exhaustion. If Chapter 3 is not designed, it may be a little difficult for those who are not familiar with the field of nanomaterials to understand.

Please clarify the meaning of 'remodeled' as its meaning is unclear. Why is M2-TAM transformation to M1-TAM remodeling? What is the transformation? Please explain it in detail.

Answer: Thanks for your comments. Remodeling means transforming or polarizing. The majority of macrophages in the tumor microenvironment exhibit an anti-inflammatory, M2-like phenotype, and the number of M2-like TAMs is associated with poor prognosis and drug resistance. M1-like TAMs are historically regarded as anti-tumor, while M2-like TAMs exhibit powerful pro-tumor functions. Moreover, TAMs suppress effective antitumor immune responses by expressing immunosuppressive molecules (IL-10, TGF-β). Macrophages change their phenotypic and functional activities according to the stimuli they receive from the TME. The tumor microenvironment can influence macrophage recruitment and polarization, giving way to these pro-tumorigenic outcomes. Therefore, to promote effective tumor therapy it is crucial to reprogram the TAMs phenotype from M2-like to M1-like by remodeling the tumor microenvironment.

Macrophage polarization refers to the activation state of a macrophage at a singular point in time, but due to the plasticity of macrophages, their polarization state is not fixed and can be altered based on the integration of multiple signals from other cells, tissues, and pathogens. Either ablation of TAMs or switching of macrophages into the antitumor M1-like phenotype results in a significant reduction in tumor growth. We have added some words to expand the 4.2 section to make it easier to understand.

References:

[1] Han S, Wang W, Wang S, Yang T, Zhang G, Wang D, Ju R, Lu Y, Wang H, Wang L. Tumor microenvironment remodeling and tumor therapy based on M2-like tumor associated macrophage-targeting nano-complexes. Theranostics. 2021 Jan 1;11(6):2892-2916.

[2] Boutilier AJ, Elsawa SF. Macrophage polarization states in the tumor microenvironment. Int J Mol Sci. 2021 Jun 29;22(13):6995.

[3] He Z, Zhang S. Tumor-associated macrophages and their functional transformation in the hypoxic tumor microenvironment. Front Immunol. 2021 Sep 16;12:741305.

[4] Zhang Q, Sioud M. Tumor-associated macrophage subsets: shaping polarization and targeting. Int J Mol Sci. 2023 Apr 19;24(8):7493.

Some NPs in Table 1 are not characterized (in column 2).

Answer: Thanks for your comments. I carefully examined the NPs in Table 1 and added some descriptions of NPs.

- The last section failed to mention the perspectives, trends, and future tasks.

Answer: Thanks for your comments. We have added more recent articles, corrected the conclusion of the manuscript, and further expanded the manuscript.

- The references are incorrectly formatted and contain some inconsistencies. There are some abbreviated journal names. Abbreviated names need dots, and the full journal names are also incorrectly spelled.

Answer: Thanks for your comments, I carefully revised the format of the references one by one, to make it more accurate.

We have revised our manuscript very carefully. If there are any questions, please let us know and we will do more revisions. We are looking forward to hearing from you.

Best wishes,

Fei Li, M. D.

Round 2

Reviewer 1 Report

Comments and Suggestions for Authors

The authors well addressed my concorns and the reivsed vesion is now more appealing

Reviewer 2 Report

Comments and Suggestions for Authors

.

Comments on the Quality of English Language

 Minor editing of English language required.

Reviewer 4 Report

Comments and Suggestions for Authors

The authors have responded satisfactorily to the reviewers' concerns and have corrected their manuscript accordingly. It is suitable for publication, although it is necessary to point out to the authors that some abbreviations in the references are still missing dots (refs. 2, 15, 16, 17, 49, 51, 52, 56, 57, 60, 63, 64, 66, 67, 70, 74, 77, 78, 79, 82, 86, 92).